# MatriCom, a single-cell RNA-sequencing data mining tool to infer cell–extracellular matrix interactions

Rijuta Lamba[1,2], Asia M. Paguntalan[3,*], Petar B. Petrov[4,*], Alexandra Naba[3,5,‡] and Valerio Izzi[1,2,3,‡]

## ABSTRACT

The extracellular matrix (ECM) is a complex meshwork of proteins forming the framework of all multicellular organisms. Protein interactions are critical to building and remodeling the ECM meshwork, while interactions between ECM proteins and their receptors are essential to initiate signal transduction. Here, we present MatriCom, a web application (https://matrinet.shinyapps.io/matricom) and a companion R package, devised to infer communications between ECM components and between different cell populations and the ECM from single-cell RNA-sequencing (scRNA-Seq) datasets. MatriCom relies on a unique database, MatriComDB, of over 25,000 curated interactions involving matrisome components to impute interactions from expression data. MatriCom offers the option to query user-generated or open-access datasets sourced from large sequencing efforts. MatriCom also accounts for specific rules governing ECM protein interactions. We illustrate how MatriCom can generate novel biological insights by building the first human kidney matrisome communication network. Last, applied to a panel of 46 scRNA-Seq datasets of healthy adult tissues, we demonstrate how MatriCom can shed light on the mechanisms of conservation and diversification of ECM assemblies and cell–ECM interactions.

KEY WORDS: Co-expression, Extracellular matrix, Matrisome, Protein–protein interactions, Network analysis

## INTRODUCTION

The extracellular matrix (ECM) is a complex and dynamic meshwork of proteins that forms the framework of all multicellular organisms (Karamanos et al., 2021; Naba, 2024). We have previously defined the 'matrisome' as the compendium of genes encoding the structural components of the ECM ('core matrisome', e.g. collagens, fibrillar glycoproteins, proteoglycans) and associated modulatory components (e.g. proteases or cross-linking enzymes that modulate ECM architecture, growth factors or morphogens that bind core ECM components) (Naba et al., 2012). This compendium comprises ~1000 genes in the mammalian genomes (Gebauer and Naba, 2020; Hynes and Naba, 2012).

Importantly, protein–protein interactions are critical for ECM protein functions. Indeed, matrisome–matrisome protein interactions that can occur intracellularly in the biosynthetic and secretory pathway or extracellularly are essential for the proper assembly of the ECM meshwork (Naba, 2024). In addition, ECM proteins exert signaling functions through interactions with transmembrane receptors. These interactions trigger molecular cascades orchestrating most cellular phenotypes, including cell proliferation and survival, stemness, differentiation and migration (Hastings et al., 2019; Hynes, 2009; Pally and Naba, 2024). As a result, alterations of ECM protein interactions compromise the structural integrity and signaling functions of the ECM and have significant consequences (Naba, 2024; Theocharis et al., 2019), as illustrated by the plethora of diseases arising from ECM gene variants (Lamandé and Bateman, 2020) or observations made in the context of cancer (Cox, 2021) and fibrosis (Walraven and Hinz, 2018). Yet, as of now, we lack the tools to probe these interactions with high throughput in the context of a tissue microenvironment and ask fundamental questions pertaining to ECM biology. For example, although we and others have previously shown that matrisome genes present cell-type-specific and organ-specific expression patterns (Nieuwenhuis et al., 2021; Tellman et al., 2022), we have yet to determine which cell populations within a tissue or organ contribute to the building of the ECM meshwork and whether similar populations do so across organs. Moreover, recent proteomic studies have revealed that the ECM of any given tissue is composed of ~100–200 distinct proteins and ~25% of these proteins are ECM regulators, i.e. enzymes involved in ECM remodeling (Naba, 2023; Shao et al., 2023), yet we do not know which cell populations express these regulators and contribute to the remodeling of the assembled ECM meshwork. Last, we have yet to build a comprehensive atlas of pairs of sender and receiver populations of ECM signals to map communications in a tissue microenvironment precisely.

The past decade has seen an acceleration of atlasing efforts aimed at mapping biomolecules with high throughput and at the single-cell resolution in health and disease conditions (Ando et al., 2020). This acceleration was made possible thanks to the development of accessible technologies such as single-cell RNA sequencing (scRNA-Seq) and also by the establishment of large consortia, including those supported by the National Institutes of Health (NIH) in the United States, such as the Human Biomolecular Atlas Program (HuBMAP), focused on the profiling of biomolecules in adult healthy tissues (Jain et al., 2023); the Cellular Senescence Network (SenNet), focused on the profiling of senescent cells across human organs (SenNet Consortium, 2022); the Human Tumor Atlas Network (HTAN), focused on cancers (Rozenblatt-Rosen et al., 2020); or organ-specific efforts such as the Kidney Precision Medicine Project (de Boer et al., 2021) or LungMap (Gaddis et al., 2024). In conjunction with these atlasing efforts, tools have been developed to leverage the wealth of data generated to infer

[1]Faculty of Biochemistry and Molecular Medicine, University of Oulu, Oulu FI-90014, Finland. [2]Faculty of Medicine, Biomedical and Internal Medicine Research (BioIM) Unit, University of Oulu, Oulu FI-90014, Finland. [3]Department of Physiology and Biophysics, University of Illinois Chicago, Chicago, IL 60612, USA. [4]Infotech Institute, University of Oulu, Oulu FI-90014, Finland. [5]University of Illinois Cancer Center, Chicago, IL 60612, USA.
*These authors contributed equally to this work

‡Authors for correspondence (valerio.izzi@oulu.fi; anaba@uic.edu)

A.N., 0000-0002-4796-5614; V.I., 0000-0002-9960-4917

protein–protein interactions and signaling events from scRNA-Seq (i.e. gene expression) datasets. Such tools include CellChat (Jin et al., 2021, 2024), NicheNet (Browaeys et al., 2020), CellTalkDB (Shao et al., 2021) and SingleCellSignalR (Cabello-Aguilar et al., 2020). These tools broadly focus on ligand–receptor interactions and rely on custom interaction databases. Interrogation of the interaction databases supporting these tools has revealed a significant under-representation of matrisome interactions, especially with regard to the core matrisome. For example, CellChatDB lists 4870 interactions involving human proteins, of which 2344 involve at least one matrisome component, but only 111 (∼2.2%) are between two matrisome components, and none are between two core matrisome components. Although ∼40% of the 12,659 interactions reported in NicheNet involve at least one matrisome component, only 3% involve one core matrisome component. Similarly, CellTalkDB compiles 3398 interactions, of which 1877 include a matrisome gene either as ligand or receptor, but only 458 (∼13%) involve at least one core matrisome component, 98 (∼2.9%) are between two matrisome components, and only six interactions are between two core matrisome components.

Importantly, matrisome–matrisome protein interactions obey certain rules. For example, functional collagens, the most abundant core ECM proteins, are triple helical proteins. Although 44 genes encode α chains of collagens in the human genome, the assembly of these chains into functional trimers is highly selective and results in 28 functional collagen proteins. Collagen protomers, formed of three collagen α chains, must assemble intracellularly before being secreted extracellularly (Naba, 2024; Ricard-Blum, 2011). Similar rules govern the assembly of trimers of laminins that are structural components of the basement membrane ECM and any other multimeric core matrisome proteins (Naba, 2024). These examples demonstrate that stipulations need to be imposed when inferring protein interactions from scRNA-Seq datasets: (1) that not all collagen chains can assemble with each other and (2) that genes encoding the individual components of functional protomers should be expressed by the same cell and not by different cells. Of note, selective chain assembly does not only apply to collagens but also to trimeric laminins (12 genes in the human genome), key structural components of basement membrane ECMs (Hohenester, 2019) or to the largest class of ECM receptor, the integrin heterodimers (Hynes, 2002).

To account for the specificities governing interactions involving matrisome components, we developed MatriCom, a web application (https://matrinet.shinyapps.io/matricom) and a companion R package (https://github.com/Izzilab/MatriCom), devised to mine scRNA-Seq dataset and infer ECM–ECM and cell–ECM communication systems. MatriCom relies on a unique database, MatriComDB, that includes over 25,000 curated interactions involving matrisome components sourced from seven interaction databases, resulting in data on 80% of the matrisome. In addition, MatriCom offers a model-maximization functionality to account for rules governing ECM interactions (e.g. constraints on homocellular versus heterocellular co-expressions or selective partner assemblies) and a ranking of the reliability of the interactions. We then illustrate the usability of MatriCom through two examples using publicly available scRNA-Seq datasets: (1) the building of the kidney matrisome communication system and (2) the integration of scRNA-Seq datasets from 46 datasets leading to the identification of ubiquitous and tissue-specific ECM communication patterns. These two examples aim to highlight how MatriCom can be used as a hypothesis-generating resource to gain novel insights into the roles of different cell populations in the building of the ECM and cell–ECM interactions and their dysregulations in the context of diseases such as cancer or fibrosis.

# RESULTS

## Construction of MatriComDB, an omnibus matrisome interaction database

To build MatriComDB, we sourced interactions from seven databases, including two ECM-focused databases – MatrixDB and MatrixDB-curated interactions of the IMEX database, and basement membraneBASE – and more generalist interaction databases – Kyoto Encyclopedia of Genes and Genomes (KEGG), BioGRID, STRING and OmniPath (Fig. 1A, Table 1; Fig. S1). In aggregate, MatriComDB comprises 26,571 unique interactions involving at least one matrisome component. Only 8.2% of these interactions are listed by more than one source database (Fig. 1A, right; Table S1A), highlighting the importance of having sourced interactions from multiple databases. Of the 26,571 interactions listed in MatriComDB, over 20,000 are between a matrisome protein and a non-matrisome protein and 6373 (∼24%) are between two matrisome proteins (Fig. 1B; Table S1A). The interactions between a matrisome protein and a non-matrisome protein include interactions with intracellular proteins (9390 interactions), proteins found at the cell surface (6518 interactions) and proteins found in the extracellular space but not the ECM (4290 interactions) (Fig. 1C; Table S1A). Of the 6373 interactions engaging two matrisome proteins, 1442 involve two core matrisome proteins, 1309 involve a core matrisome and a matrisome-associated protein (examples of these interactions include interactions between a collagen substrate and a collagen-crosslinking enzyme such as lysyl oxidase), and 3622 are between two matrisome-associated proteins (Fig. 1D; Table S1A). The contribution of the seven source databases to these different types of interactions included in MatriComDB is provided (Fig. S2).

With the matrisome in focus, MatriComDB includes interactions for the proteins encoded by nearly 6000 genes, of which 995 are matrisome genes (Fig. 1E, left; Table S1B), covering 81% of the in-silico-predicted matrisome and all categories of matrisome components, including structural core matrisome proteins, such as collagens, proteoglycans and ECM glycoproteins, as well as matrisome-associated proteins, such as ECM-affiliated proteins, ECM regulators and ECM-bound secreted factors (Fig. 1E, right; Table S1B).

### The MatriCom application

MatriCom is made available to users as an online web application (available at https://matrinet.shinyapps.io/matricom/) and an offline version served through the MatriCom package (https://github.com/Izzilab/MatriCom). The only differences between the two versions are the availability of preprocessed open-access (OA) sample datasets, which are only available on the online version, and the size of permitted uploads (maximum 1 GB online, unrestricted offline). We will use the online version to illustrate MatriCom functionalities throughout this paper.

### MatriCom input

Users can upload their own datasets from Seurat (Hao et al., 2021) or Bioconductor's Single Cell Experiment (SCE) pipelines in .rds or .qs format, or from ScanPy/Loom in .h5ad format via the 'Data Input section'. In the online version, users can also select a dataset from a list of OA sample datasets available from the Tabula Sapiens OA collection (24 healthy human organ datasets; Table S2A), The Human Protein Atlas (THPA) (31 healthy human tissue datasets; Table S2B) and other studies (four organs; Table S2C).

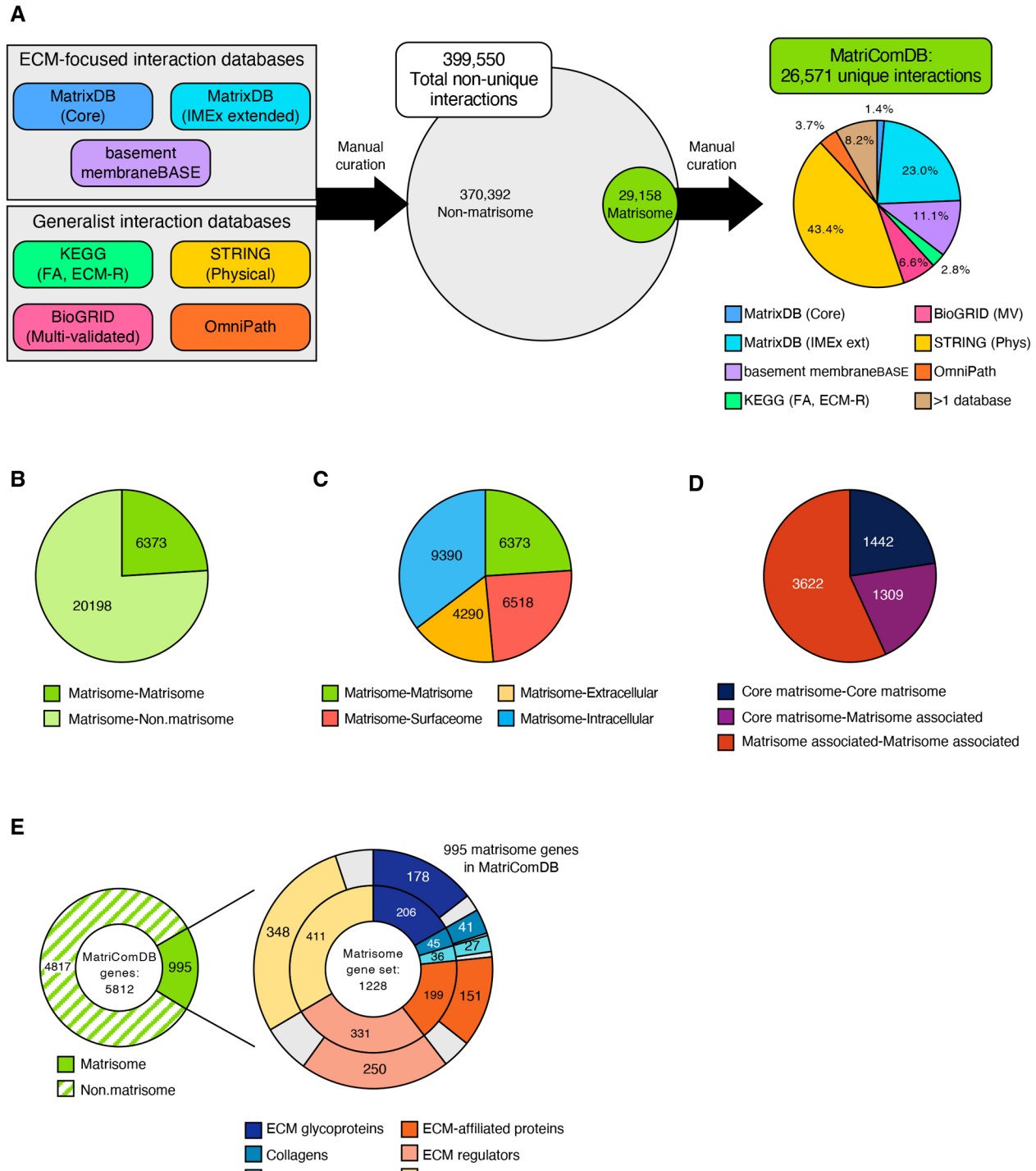

Fig. 1. Building MatriComDB: an omnibus database of matrisome protein interactions. (A) Schematic representation of the workflow devised to construct MatriComDB, an omnibus matrisome interaction database. Left: interactions were retrieved from three ECM-focused and four generalist interaction databases. Middle: Venn diagram represents the proportion of interactions retrieved involving no matrisome component (gray) or at least one matrisome protein (green). Only interactions involving at least one matrisome protein were included in MatriComDB. Right: pie chart represents the contribution of each database to the unique interactions composing MatriComDB. See also Table S1A. (B) Pie chart represents the number of unique interactions in the omnibus database involving two matrisome proteins (green) or one matrisome and one non-matrisome protein (light green). (C) Pie chart represents the number of unique matrisome protein–matrisome protein (green), matrisome protein–surfaceome protein (red), matrisome protein–extracellular but non-matrisome protein (yellow), and matrisome protein–intracellular protein (blue) interactions in MatriComDB. The following hierarchy is applied to characterize gene products that may localize to multiple compartments: Matrisome>Surfaceome>Extracellular (Non-matrisome)>Intracellular. (D) Pie chart represents the number of unique interactions between two core matrisome proteins (dark blue), a core matrisome protein and a matrisome-associated protein (purple), or two matrisome-associated proteins (red) in MatriComDB. (E) Left: donut chart represents the number of matrisome and non-matrisome genes represented in MatriComDB. Right: inner donut represents the composition of the combined mouse and human matrisome gene set; outer donut displays the coverage of matrisome genes in MatriComDB for each category of matrisome components. See also Table S1B. ECM, extracellular matrix; ECM-R, ECM receptor; FA, focal adhesion; KEGG, Kyoto Encyclopedia of Genes and Genomes; MV, multi-validated.

**Table 1. Source database curated to build the omnibus matrisome interaction database**

| Source database (subset) | Description | URL | Reference | Number of interactions retrieved | Number of interactions included in MatriComDB |
|---|---|---|---|---|---|
| MatrixDB (Core) | ECM-focused database of experimentally validated interactions involving ECM proteins, proteoglycans and polysaccharides. Provides additional web-based tools to construct and analyze matrix protein interaction networks. | https://matrixdb.univ-lyon1.fr | Clerc et al., 2019 | 758 | 746 |
| MatrixDB (IMEx ext) | Extension of the 'Core' MatrixDB dataset that includes additional interactions provided and curated by the IMEx consortium. | http://www.imexconsortium.org | Orchard et al., 2012 | 10,3711 | 7367 |
| basement membraneBASE | Knowledgebase providing experimental data on gene expression and protein localization in basement membranes during development, adulthood and disease, and across animal species. | https://www.bmbase.manchester.ac.uk | Jayadev et al., 2022 | 3982 | 3534 |
| KEGG PATHWAY (FA, ECM-R) | Integrated resource of manually curated datasets constituting 16 individual databases, including the PATHWAY database from which the FA (hsa04510) and ECM-R (hsa04512) interaction datasets were retrieved. | https://www.kegg.jp/kegg/pathway.html | Kanehisa et al., 2023 | 2020 | 1179 |
| BioGRID (MV) | Repository of physical and genetic interactions curated from high-throughput screens and individual studies. Multiple datasets of physical interactions supported by experimental evidence from multiple publication sources and/or validated in multiple experimental systems. | https://www.thebiogrid.org | Oughtred et al., 2020 | 129,373 | 2491 |
| STRING (Physical) | Database of known and predicted protein–protein interactions derived from computational predictions, high-throughput experiments, co-expression data and curation of other databases. Physical subnetwork datasets ('Physical') compile associations between proteins part of the same physical complex. | https://www.string-db.org | Szklarczyk et al., 2023 | 122,655 | 12,191 |
| OmniPath | Knowledgebase of intra- and inter-cellular signaling integrating data from 100+ resources into five individual databases focused on molecular signaling networks, enzyme–PTM relationships, protein complexes, protein annotations and intercellular communication. | https://www.omnipathdb.org | Türei et al., 2021 | 37,051 | 1650 |

ECM, extracellular matrix; ECM-R, ECM receptor; IMEx, International Molecular Exchange; KEGG, Kyoto Encyclopedia of Genes and Genomes; MV, multi-validated; PTM, post-translational modification.

## Query parameters

### Cell annotations

Users can select the metadata column to be used to label cell identities from their dataset using the first dropdown menu of the Query Parameters tab. For OA sample datasets, both original and standardized cell identities are available (see Materials and Methods).

### Expression level and population thresholds

Once cell identity labels are set, users should choose the proportion of cells (per label) that express any gene involved in matrisome communication and the minimum mean level of gene expression to be met to consider a gene as 'expressed'. Of note, lower values encourage more granular results at the expense of computational time, while higher values select for the most abundant and more stable matrisome features of a dataset but mask more subtle, and potentially important, communications systems in less abundant cell types.

## Post-run filters

MatriCom implements a set of filters to account for specific features of ECM biology that might affect the precise inference of communication systems. In particular, we have devised options to (1) maximize the output of a MatriCom model by only listing the most reliable communicating pairs in case of duplicates identified in multiple source databases of different reliability; (2) exclude 'impossible' pairs between different cells, such as collagen protomers which strictly assemble within the same cells (Naba, 2024); (3) remove pairs where both elements are the same gene (e.g. the same collagen chains), as the likeliness of these would inflect on the recovery of heterotypical ones; and (4) filter the results based on the reliability score of each pair, the type of communication (between the same or different cell type, or both), and the subcellular location of the gene product (matrisome, cell surface, extracellular and intracellular space). Of note, these filters are activated by default but can be deactivated upon relaunching a MatriCom run.

## MatriCom output

At completion of the analytical operations and initial filtering, a rich output is returned to users in both graphical and tabular formats in a few seconds to minutes, depending on the dataset size. To facilitate navigation, results are grouped into tabs related to a specific functional part of the output:

### Communication network

This tab provides users with a 'Global Communication Cluster Map' (Fig. 2B) derived from the dataset queried. The map represented as a bubble plot reports the number of matrisome communication pairs for each cell population pair. The map is interactive: users can click on any bubble to restrict the results and downstream analyses to a particular population pair.

In the same tab, users will have access to a bar graph summarizing the number of matrisome interactions ('matrisome–matrisome' or 'matrisome–non-matrisome'; Fig. 2C) and a bubble plot reporting the number of matrisome interactions involving the different categories of matrisome components (Fig. 2D). Last, in this tab, users have access to the full output in a tabular format. The table lists, for each communication pair, the partners involved (gene symbols), the type of communication ('matrisome–matrisome' or 'matrisome–non-matrisome'), the identities of the cell populations involved, the proportion of each cell population expressing the genes of the communicating pair, and the mean gene expression levels of the two partners (Fig. 2E).

### Network influencers

In this tab, MatriCom results are processed into a non-directed, acyclic graph, and the relative importance of any graph node (any gene) in directing 'traffic' towards its neighboring nodes is calculated, thus enabling users to identify the most noticeable (and potentially more actionable or interesting) genes within their results (Fig. S3A).

### Enrichment analysis

In this tab, MatriCom provides users with *ad hoc* enrichment analysis against a set of manually curated matrisome-specific signatures (Fig. S3B).

### Case study: building the human kidney matrisome communication network

To illustrate the usability of MatriCom, we reanalyzed a previously published scRNA-Seq dataset of the healthy adult human kidney (https://www.ebi.ac.uk/biostudies/studies/S-SUBS7) using the default query parameters, settings and post-run filters. MatriCom analysis returns a total of 12,528 matrisome communications established by 793 distinct pairs established between the 33 cell types represented in the original sample dataset (Fig. 3A; Table S3A). The full MatriCom output for this dataset with original cell identity annotation, including network influencer and enrichment analyses, are provided in Table S3A–C. To demonstrate the cell annotation functionality embedded in MatriCom, we also provide users with the same dataset and query parameters but generated by selecting the 'Census' reannotation option (Table S3D–F).

Communications between genes expressed by the same population – i.e. homocellular pairs – account for only 6.5% of the full network, while most communications, 93.5%, are established by heterocellular pairs (Table S3A). Analysis of the full communication network reveals that non-matrisome–matrisome pairs comprise the majority (78.4%) of all communications pairs (Fig. 3A). Of the pairs involving two matrisome components, only 333 (2.7%) involve two core matrisome components (Fig. 3A). Yet we know that core matrisome–core matrisome component interactions are foundational to ECM assembly. This observation reflects, to a certain extent, the content of MatriComDB, which is biased toward non-matrisome and matrisome-associated proteins and highlights the need to develop tools to identify interactions between core matrisome components (see Discussion).

To determine the degree to which individual populations contribute to the kidney matrisome communication network, we determined a number of expressions per population (Fig. S4A), which we defined as the number of times a given population appears in the MatriCom communication network table as either Population1 or Population2. Together, seven populations make up over 50% of all expressions: distinct proximal tubule 1, connecting tubule, epithelial progenitor cell, pelvic epithelium, principal cell and fibroblast (Fig. 3B; Table S4A). Because cell count in the original dataset varies between populations (Fig. S4B), we corrected for population size and found that, despite comprising less than 0.25% of the original sample dataset, fibroblasts are the largest contributors to the kidney matrisome communication network (Fig. 3B; Table S4A). Interestingly, the relative contribution of expression by fibroblasts is nearly twice that of distinct proximal tubule 1 cells, the next largest contributor (Fig. 3B; Table S4A). Because fibroblasts are the primary cell type responsible for ECM deposition in most tissues (Naba, 2024), we next sought to focus on matrisome communications established by fibroblasts.

### Fibroblast-specific communications

We found that 1663 communication pairs (corresponding to ~13% of the kidney matrisome communication network) involved fibroblasts (Fig. 3C; Table S4B). These include 618 (37.2%) matrisome–matrisome pairs and 1045 (62.8%) non-matrisome–matrisome pairs (Fig. 3C). Heterocellular pairs account for over 96% of the fibroblast network, such that less than 4% of communications are between fibroblasts (Table S4B). Only 143 (8.6%) communication pairs involve two core matrisome components (Fig. 3C; Table S4C–E, Fig. S4C), yet the proportion of communications between two core matrisome genes in the fibroblast network is over threefold higher than that of the full kidney network (Fig. 3A,C).

To evaluate the contribution of the different cell populations to the fibroblast-specific matrisome communication network, we computed all population pairs and determined the combined (e.g. fibroblast–partner and partner–fibroblast) communication frequency per partner (Table S4C). We found that the top five contributing populations – distinct proximal tubule 1, principal cell, connecting tubule, pelvic epithelium and epithelial progenitor cell – establish more non-matrisome–matrisome communications with fibroblasts than matrisome–matrisome communications (Fig. 3D; Table S4C, Fig. S4D).

### Fibroblast ECM–receptor communications

Overall, nearly half (46.6%) of all fibroblast communications involve at least one core matrisome gene, including 41.9% of matrisome-matrisome pairs and 81.8% of matrisome–surfaceome pairs (Table S4B–D). Therefore, we further sought to explore the subnetwork of communication pairs between ECM and ECM receptors involving fibroblasts. To do so, we extracted from the global fibroblast network the communication pairs represented in the KEGG ECM–receptor interaction pathway (hsa04512) for which fibroblasts are the 'sender' population – i.e. express the genes encoding ECM ligands (Table S5A). The fibroblast ECM-receptor subnetwork comprises 408 communication pairs. Of these, 324 (79.4%) are non-matrisome–matrisome pairs and 84 (20.6%) are matrisome–matrisome pairs (Table S5).

Inspection of the subnetwork revealed that ten unique ligand genes and 16 unique receptor genes account for all pairs of the fibroblast ECM–receptor communication subnetwork (Fig. 4A,B; Table S5B). All ten ligand genes belong to the core matrisome, including five ECM glycoprotein genes (*FN1*, *THBS1*, *SPP1*, *LAMB2* and *LAMC3*) and five collagen genes (*COL1A1*, *COL1A2*, *COL6A1*, *COL6A2*,

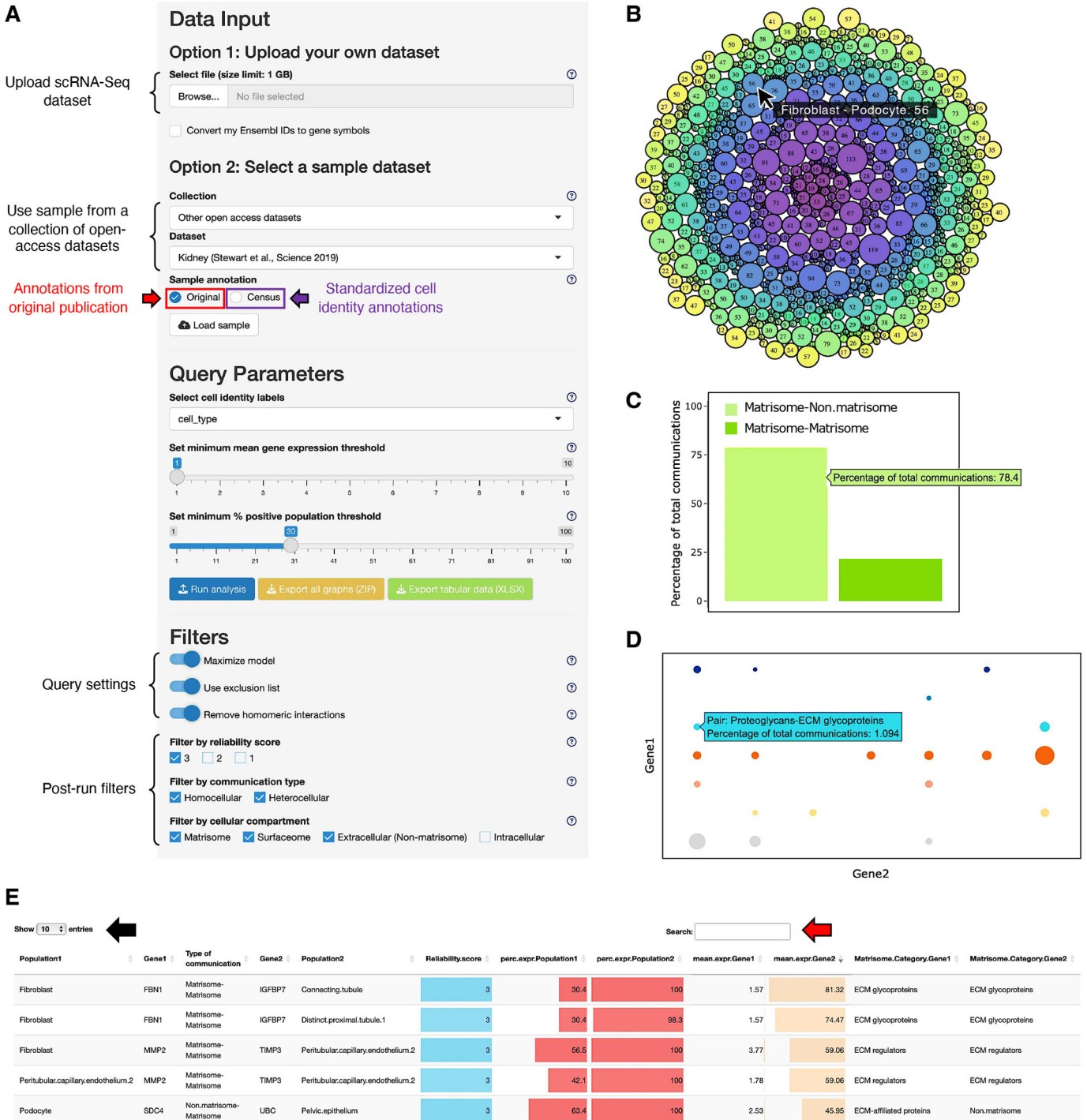

**Fig. 2. MatriCom user interface.** (A) Data input and query parameter options users can select are shown on the left. After running an analysis (blue button), users can refine the output of their search by using post-query filters and use export buttons to download all graphs (yellow button) and tabular data (green button). (B) The 'Global Communication Cluster Map' represents each unique pair of communicating cell populations as a circle. The circle size is proportional to the number of communications for each cell population pair. Hovering over individual circles will reveal details about cell population pairs and numbers of communication pairs defined as pairs of genes encoding proteins interacting based on MatriComDB. (C) The 'Communication Pairs' bar chart represents the percentage of matrisome–matrisome (green) or matrisome–non-matrisome (light green) communication pairs in the MatriCom analysis output. Hovering over individual bars will show data labels. Results are dynamically adjusted to the selection made in the 'Global Communication Cluster Map'. (D) The 'Matrisome Pairs' bubble plot represents the distribution of communication pairs with respect to the classification of genes involved according to the matrisome nomenclature. The bubble size is proportional to the number of communications. Hovering over individual bubbles will show data labels. (E) The MatriCom output table displays the complete list of communicating genes (i.e. genes for which protein products are known to interact) and population pairs. Additional columns provide specific information on communication type (matrisome–matrisome or matrisome–non-matrisome), reliability score, the percentage of positive population for each communication pair, the mean gene expression of each gene of a pair in each cell population, and matrisome categories of the communicating genes. Users can select the number of entries shown per page (black arrow) or query specific genes and/or populations using the search bar (red arrow).

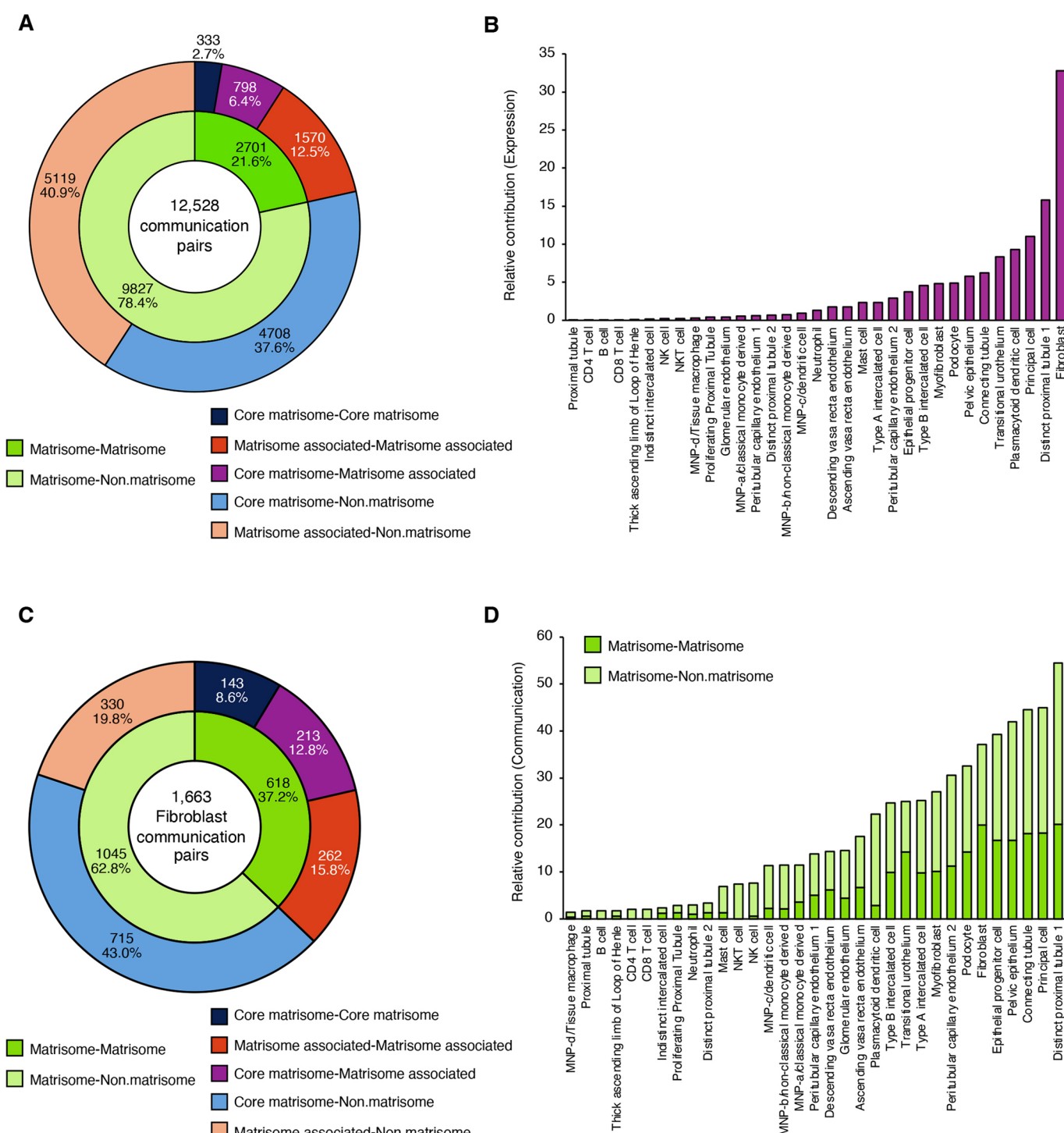

Fig. 3. Communication networks of the kidney matrisome. To illustrate the usability of MatriCom, we reanalyzed a previously published single-cell RNA-sequencing (scRNA-Seq) dataset of the adult human kidney (https://www.ebi.ac.uk/biostudies/studies/S-SUBS7). See also Tables S3 and S4. (A) Donut chart displays the distribution of communication pairs in the subset of cell population pairs involving fibroblasts with respect to communication type (inner chart) and nature of matrisome components (outer chart). (B) Bar chart represents the contribution of each cell population partnering with fibroblasts to the kidney matrisome communication network relative to the population size in the original scRNA-Seq dataset. (C) Donut chart represents the relative contribution of each cell population partnering with fibroblasts to the matrisome–matrisome communication network (inner donut chart), categorized by matrisome division of the communicating genes (outer donut chart). (D) Bar chart represents the relative contribution of each cell population partnering with fibroblasts to the matrisome–non-matrisome communication network, categorized by the matrisome division classification of the communicating genes.

COL6A3) (Fig. 4A; Table S5B). Of the 16 receptor genes, two are the ECM-affiliated transmembrane proteoglycans syndecans 1 (SDC1) and 4 (SDC4) (Fig. 4B; Table S5B). Communications involving genes encoding integrins, the main class of ECM receptors (Campbell and

Humphries, 2011), comprise 62.0% of the fibroblast ECM–receptor network, while those involving other cell surface receptor genes make up the other 38.0% (Table S5B). ITGB1, which encodes the β1 integrin subunit, participates in 130 (31.9%) fibroblast ECM–receptor

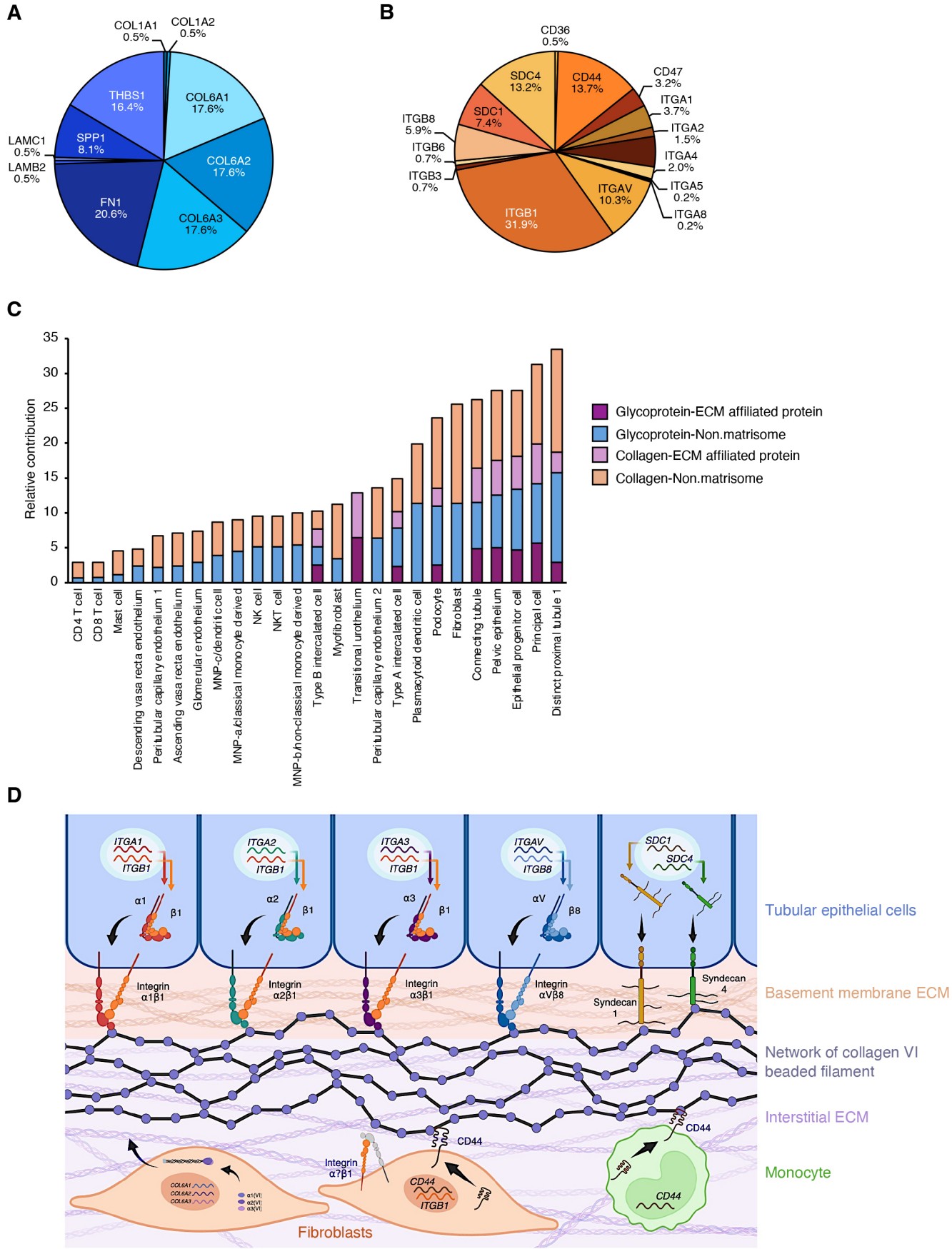

**Fig. 4.** See next page for legend.

**Fig. 4. Subset of ECM ligand–ECM receptor communication pairs involving fibroblasts in the kidney matrisome network.** (A,B) Pie charts represent the proportion of communications in the fibroblast ECM–receptor communication network that involve each ligand gene (A) and receptor gene (B). (C) Bar chart represents the relative contribution of each receiving population to the fibroblast ECM–receptor communication network, stratified by classification of the communicating genes into the different matrisome categories. (D) Schematic depicts collagen VI–receptor interactions identified by MatriCom in the fibroblast communication network. Fibroblasts express three genes – COL6A1, COL6A2 and COL6A3 – encoding the α1, α2 and α3 chains of collagen VI, respectively. These chains assemble intracellularly to form a functional collagen VI trimer termed protomer. In the cytoplasm, protomers further assemble into dimers and then tetramers, which are secreted and assembled into characteristic beaded filaments at the interface of the basement membrane ECM and the interstitial ECM. Interactions between collagen VI and its known receptors (KEGG:hsa04512) expressed by renal epithelial tubular cells, monocytes and fibroblasts are depicted. Created in BioRender by Naba, A. (2025). https://BioRender.com/q79z585. This figure was sublicensed under CC-BY 4.0 terms.

communications. *ITGAV*, which encodes the αV integrin subunit, is involved in 42 of the 408 pairs, constituting a little over 10% of all fibroblast ECM–receptor communications (Fig. 4B). Additionally, *CD44*, *SDC1* and *SDC4* account for 56 (13.7%), 30 (7.4%) and 54 (13.3%) pairs, respectively (Fig. 4B).

We next evaluated the contribution of fibroblast–receiver population pairs to the fibroblast ECM–receptor communication network (Table S5C; Fig. 4C) and found that distinct proximal tubule 1 cells are the largest contributors, followed by principal cells, epithelial progenitor cells, pelvic epithelium and fibroblasts (Fig. 4C). Homocellular pairs contribute only collagen–non-matrisome and ECM glycoprotein–non-matrisome communications, such that no expression of *SDC1* or *SDC4* receptor genes by fibroblasts within the ECM–receptor network was detected by MatriCom analysis at the default thresholds (Fig. 4C).

Notably, fibroblasts express all three genes – *COL6A1*, *COL6A2* and *COL6A3* – of the [α1(VI)α2(VI)α3(VI)] alpha-chain trimer, the functional protomer that assembles extracellularly to form collagen VI beaded filaments at the interface of the basement membrane ECM and the interstitial ECM (Cescon et al., 2015; Ricard-Blum, 2011). Together, *COL6A1*, *COL6A2* and *COL6A3* participate in over half, 52.%, of all fibroblast ECM–receptor communications returned by MatriCom (Fig. 4A). Of the 216 communications involving one of these three collagen VI genes, integrins make up 135 (62.5%) of receptor genes, while non-integrin cell surface receptor genes participate in the other 81 (37.5%) communication pairs (Table S5D). Of the kidney tubule cell populations, distinct proximal tubule 1 cells express three distinct sets of integrin genes encoding heterodimers that interact with collagen VI trimers – *ITGA1–ITGB1*, *ITGA3–ITGB1* and *ITGAV–ITGB8* – while connecting tubule and principal cells are the only populations with detectable expression of genes encoding the *ITGA2–ITGB1* heterodimer, in addition to *ITGAV–ITGB8* (Table S5F). Nine of the 12 immune cell populations express the *CD44* receptor gene and establish communications with all three collagen VI ligand genes (Table S5E). Altogether, MatriCom results can be leveraged to infer the interaction network of any given matrisome component (here, the collagen VI molecule) with its receptors in the context of any organ's cell population make-up (Fig. 4D).

## Pan-organ analysis reveals ubiquitous and tissue-specific ECM communication systems

The ECM is foundational to tissue organization, development, growth, maintenance and repair (Naba, 2024). We thus hypothesized

that parts of its communication systems may be under conservative pressure and reused by different organs facing similar biochemical demands. To test this, we investigated the expression of stringent matrisome communication pairs or 'patterns' in the entire OA dataset collection made available in MatriCom.

The MatriCom analysis was run with default parameters (mean average gene expression of 1, mean percentage positive population of 30 and reliability of 3), combining and comparing results from ~680,000 cells across the 24 tissues of the Tabula Sapiens collection and 430,000 cells across the 22 tissues of the Census-reannotated THPA collection (Fig. 5A; Table S6). To capture strong signals of pattern conservation, we then selected a minimal set of pairs that were expressed in at least 50% of all organs and tissues from Tabula Sapiens and that were also present in THPA (Fig. 5B). The 113 resulting patterns involved approximately the same number of genes in the core and associated divisions of the matrisome and non-matrisome (typically, cell receptors) (Fig. 5C), and covered all combinations of matrisome-to-matrisome categories (Fig. 5D).

In keeping with the fundamental contribution of structural cells (fibroblasts, epithelial and endothelial cells) to the total ECM output of tissues, we found that these conserved matrisome patterns largely connected stromal cells to epithelial and endothelial compartments, although all combinations are represented, with a significant set of pairs mediating connections between structural and immune cells and some even impacting on rare cells such as stem cells in Tabula Sapiens (Fig. 5E,F; Table S7A–C).

Previous reports have demonstrated that pairs and even sets of genes are often reused by multiple, different cell types to regulate diverse physiological processes in both health and disease, showcasing the versatility of gene regulatory networks, where specific combinations of genes can be co-opted for distinct functions depending on the cellular context (Fazilaty and Basler, 2023). In line with these reports, we found that the matrisome patterns show high pleiotropy and reuse, as we found them often and in multiple cell types within and across organs (Fig. S6). In particular, we noticed trends that connect the mosaic of pattern diffusion across cells and tissues with high-order functional roles of the matrisome (Fig. 6A,B), with differences in the balance of reuse versus specialization that depend largely on the functions of the given pair. For example, the interaction between collagen VI (*COL6A1*) and the CD44 receptor (*CD44*) is fundamental for cell adhesion, migration and survival, and influences processes such as tissue repair, immune response and maintenance of cell homeostasis (Cescon et al., 2015). Thus, it is not surprising to find this pair being reused by all cellular compartments but the stem cells (Fig. S6C). In contrast, the interaction between selectin-P (*SELP*) and collagen XVIII (*COL18A1*) (Izzi et al., 2020) is restricted to the endothelial and the immune compartment (Fig. S6C), as expected for the highly specialized roles of both *SELP* and *COL18A1* in vascular biology, angiogenesis and leukocyte adhesion to blood vessel walls (Dupas et al., 2024; Izzi et al., 2020).

Last, we sought to identify transcription factors (TFs) potentially regulating the co-expression of genes involved in matrisome communication pairs and patterns identified above. To do so, we interrogated two independent databases compiling transcription factor targets, TF2DNA and TFTG (Zhou et al., 2024), and identified 56 TFs as potential regulators of matrisome communication patterns, i.e. as capable of regulating both genes of a communication pair and common to both databases (Fig. S7A, Table S8). We observed that multiple TFs already reported to govern ECM transcription in health and disease are among the top hits [e.g. *EGR1*, the androgen receptor (*AR*) and *FLI1*; Fig. S7B]. Further grouping of TFs into TF families (Lambert et al., 2018) revealed enrichment of *C2H2* zinc finger TFs

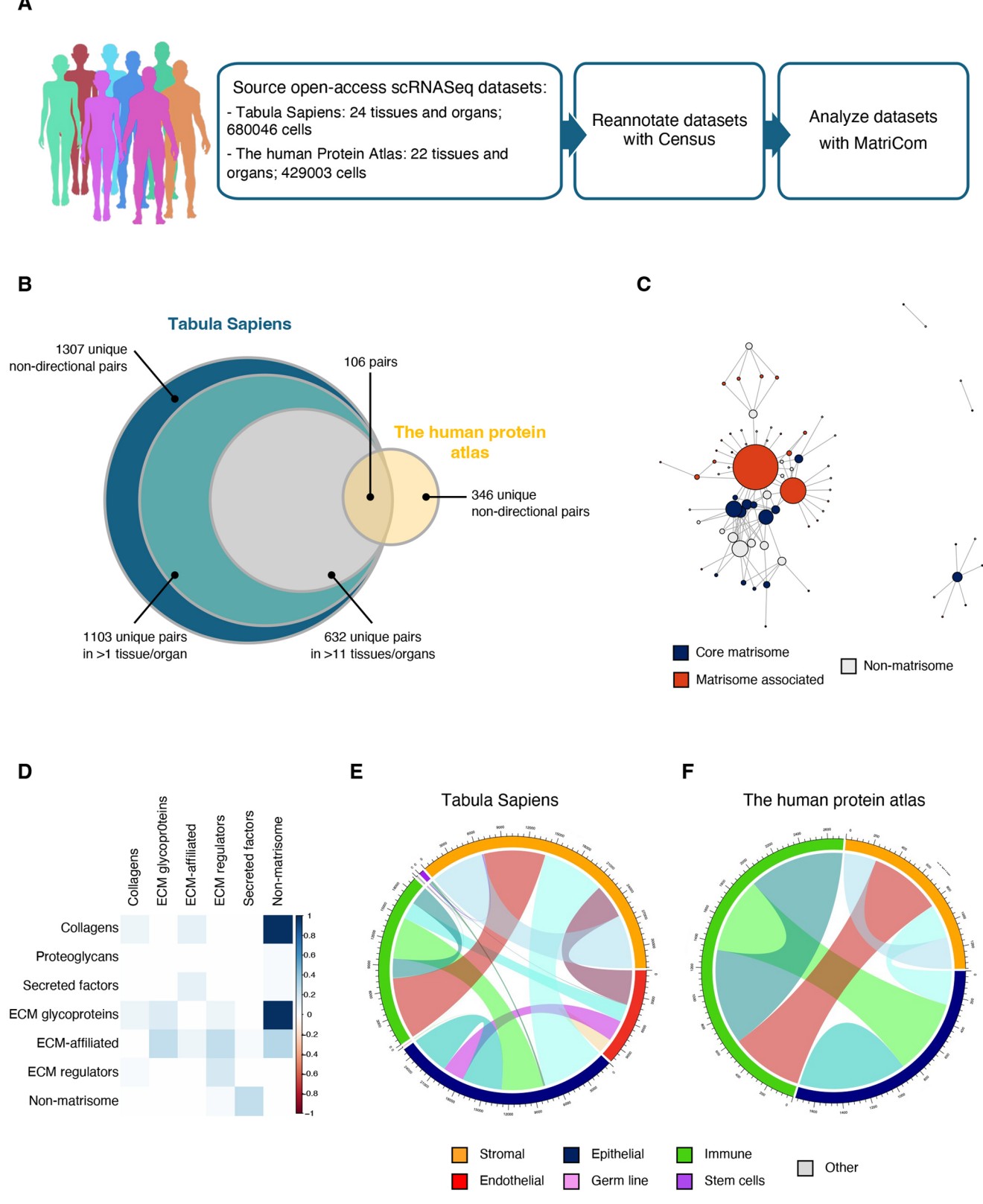

**Fig. 5.** See next page for legend.

(Lambert et al., 2018), basic helix-loop-helix (bHLH) TFs (Jones, 2004) and nuclear receptors (Lambert et al., 2018) (Fig. S7C; Table S8). As observed at the matrisome gene expression level (see above), we also identified a trend towards the functional

specialization of TFs with respect to the cell compartment expressing the pairs, in line with the fundamental role of TFs in directing and maintaining cell-of-origin identity. Specifically, we identified three TF clusters (cluster 1, comprising SP3, ETV4 and NR4A1; cluster 2,

**Fig. 5. Pan-organ analysis of matrisome communication systems.**
(A) scRNA-Seq datasets were obtained for the Tabula Sapiens and The
Human Protein Atlas open-access (OA) cohorts and analyzed using
MatriCom with default parameters. To allow inter-study analysis, cells from
The Human Protein Atlas datasets were assigned identities compatible with
Tabula Sapiens using Census (see Materials and Methods). Tabula
Sapiens, having greater depth, was chosen as the primary discovery cohort,
with The Human Protein Atlas being used to validate findings. (B) Venn
diagram depicts the number of unique non-directional pairs identified in each
dataset. We further focused on pairs discovered in at least 50% of the
available tissues from Tabula Sapiens and also present in The Human
Protein Atlas. The resulting 113 communications pairs are referred to as
patterns. (C) Network depicts the interactions between the 113 patterns.
Note the similar amount of core matrisome, matrisome-associated and
non-matrisome nodes, and the high degree of matrisome-associated nodes
(represented by the size of the orange circles), which suggest that patterns
cover a significant number of communications involving matrisome
components. (D) Matrix chart depicts the classification of partnering genes
composing the 113 patterns as matrisome (e.g. ECM glycoproteins,
collagens, ECM-affiliated proteins, ECM regulators, secreted factors) or
non-matrisome components. (E,F) Chord diagrams illustrate the relationship
between population types (stromal, epithelial, endothelial, immune, stem,
germline or other) and the matrisome communication patterns identified in
the Tabula Sapiens dataset (E) and The Human Protein Atlas dataset (F).

comprising SMAD4, SOX2 and FOXM1; and cluster 3, comprising
ARNT, GATA1, TFAP2C, NFKB1, FOXP3 and HOXB7) driving
the expression of genes of different matrisome communications pairs
and patterns (Fig. S7D). Interestingly, most of these TFs were already
reported to govern ECM expression in healthy tissues and cancers
(Chiquet et al., 2009; Izzi et al., 2019; Trojanowska, 2000), suggesting
that ECM regulation at the transcriptional level is strongly cell-type-
dependent and not greatly affected by network rewiring.

Altogether, our data demonstrate how imputing interactions from
expression data can shed light on the mechanisms of conservation
and diversification of ECM assemblies and cell–ECM interactions
that result in imparting unique functions to each organ.

## DISCUSSION
MatriCom is a one-of-a-kind tool with a framework designed to
capture the rules governing the interactions involving ECM and
ECM-related proteins with each other, and with cell surface
receptors, and infer these interactions from scRNA-Seq datasets.
The datasets used to illustrate the usability of MatriCom were
generated from healthy tissue samples, but we envision that, applied
to datasets generated on disease-relevant samples, MatriCom will
help uncover altered matrisome and cell–matrisome communication
systems and patterns and help advance our understanding of the
roles of the ECM in disease.

MatriCom output relies on the robustness of MatriComDB,
a newly developed database of matrisome protein interactions.
Although MatriComDB has been meticulously curated from diverse
sources, it is worth noting that not all offer stringent experimental
validation or satisfy the technical criteria necessary for identifying
matrisome interactions. To address the potential for erroneous
results, we introduced the concept of source reliability. By default,
MatriCom is configured to launch at the highest reliability level, set
at 3, ensuring that all identified communication pathways are
supported by robust experimental evidence. MatriCom users are
therefore encouraged to carefully consider the trade-off between
reliability and yield. The assignment of reliability scores to each
source reflects the current state of the art in matrisome-focused and
broader biological interaction databases and will be subject to future
adjustments as sources are updated or as interaction data from
additional databases are included. Consequently, interactions

deemed to have low reliability today may be reassessed and
upgraded in the future. Of note, the current content of MatriComDB,
which is biased toward non-matrisome and matrisome-associated
proteins, also highlights the need to develop tools to identify
interactions between core matrisome proteins, a persisting challenge
owing to the high insolubility of core matrisome components (Bains
and Naba, 2024; Naba, 2023).

Imputing matrisome communications from scRNA-Seq data
presents a significant challenge, one that current computational
tools can only partially address. As demonstrated, matrisome gene
counts are among the lowest in typical scRNA-Seq datasets
(Fig. S5), leading to a higher prevalence of zero counts compared
to genes encoding components of the intracellular proteome or of
the surfaceome.

Matrisome genes are, therefore, less likely to be comprehensively
captured by algorithms that rely on genetic co-expression patterns to
represent communications. In other words, matrisome genes will tend
to be under-represented in communication analyses. For example,
prior knowledge has shown that fibroblasts express a larger number of
collagen genes, including the fibrillar collagens III and V (encoded
by *COL3A1* and *COL5A1–3*, respectively) not found in our reanalysis
of the OA kidney. The dissemination of higher-resolution techniques
and velocity assessment should help capture under-represented
genes, but, at the moment, the overall low expression of matrisome
genes remains a significant challenge preventing a more
comprehensive definition of ECM communication networks.

Last, the spatial dimension of physical matrisome interactions
remains largely unmodeled and, to date, unmodelable. Unlike soluble
ligands, many ECM components are localized to specific regions of
the pericellular or intercellular space, stabilized by interactions with
cell surface receptors and other matrisome constituents. In addition,
and as discussed, some interactions between different chains of ECM
protomers can exclusively take place within a given cell and between
defined partners. Although the precise localization of these elements
is critical for ECM organization (Naba, 2024) and multicellularity
(Bich et al., 2019; Hynes, 2012; Rokas, 2008), the actual physical
distances between communicating matrisome elements are currently
unknown. We anticipate that the broad adoption of methods such as
spatially resolved transcriptomics (Marx, 2021) and proteomics
(Method of the Year, 2024: spatial proteomics, 2024) will help fill
this gap and add a new degree of precision to our understanding of the
extracellular space and the ECM.

## MATERIALS AND METHODS
### Construction of MatriComDB, the omnibus matrisome interaction database
MatriComDB was assembled by manual curation and compilation of the
interactions between matrisome components ('matrisome–matrisome') and
between matrisome components and cellular components ('matrisome–
non-matrisome') across seven databases.

To guide curation, we used the latest list of matrisome genes available for
download on the Matrisome Project website (https://matrisome.org) across
all our tools, such as MatrixDB (Clerc et al., 2019) and Matrisome
AnalyzeR (Petrov et al., 2023) to ensure standardization.

The following databases were interrogated in February 2024 (see also
Table 1 and Table S1):
- MatrixDB (http://matrixdb.univ-lyon1.fr/) core collection ['Core
MatrixDB dataset', PSI-MI TAB 2.7 file (Clerc et al., 2019)]
augmented with two published interactomes: 'Interaction network of
the pro-peptide of lysyl oxidase' (Vallet et al., 2020) and 'Interaction
network of the four syndecans' (Gondelaud and Ricard-Blum, 2019).
Interactions were subset to human and mouse (taxid:9606 and
taxid:10090, respectively) and to UniProt and gene symbol

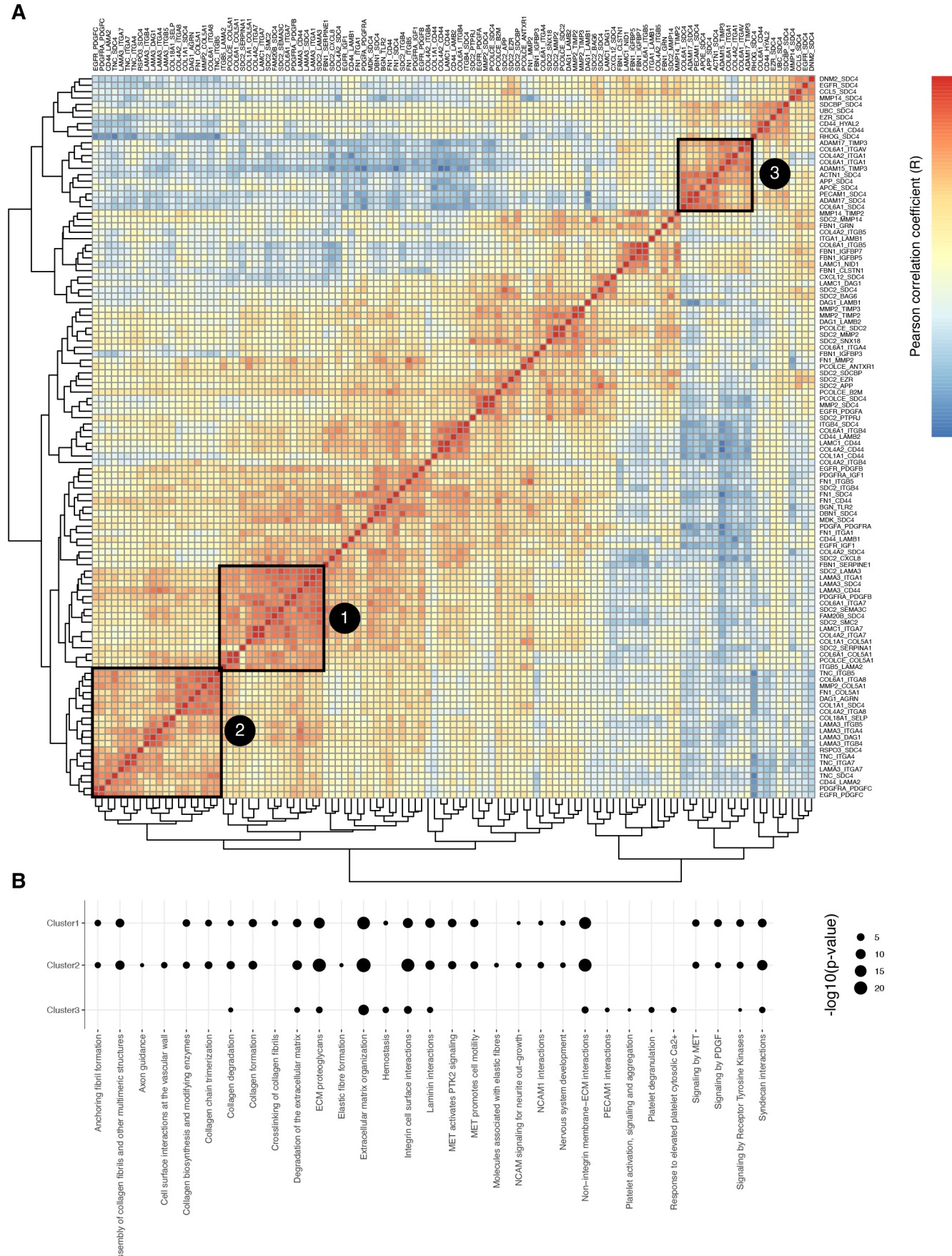

**Fig. 6.** See next page for legend.

**Fig. 6. MatriCom identifies conserved and unique matrisome communication patterns across human tissues and organs.** (A) Heat map represents the correlation between communication pairs across organs and tissues and identifies patterns of similarity or clusters. Exemplary clusters were identified by visual inspection and greedy modularity optimization of the correlation matrix after thresholding for strong correlation (Pearson $R>0.7$) only. (B) Reactome enrichment analysis of three clusters identified in A. Results are reported as $-\log10(P\text{-value})$.

identifiers only, for a total of 308 binary interactions from the core collection, 329 for the syndecan interactome and 27 for the lysyl oxidase interactome. Finally, we integrated the interactome of membrane collagens (Professor Sylvie Ricard-Blum, University Lyon I, France, personal communication), reporting 111 interactions for a total of 758 unique interactions. These were finally filtered against the matrisome gene list (see above) to ensure that at least one interactor of each pair is a matrisome component, bringing the augmented core MatrixDB collection to a total of 746 interactions.

- MatrixDB – IMEx extended collection ['IMEx extended MatrixDB dataset', PSI-MI TAB 2.7 file (Clerc et al., 2019)]. Interactions were subset to human and mouse (taxid:9606 and taxid:10090, respectively) and to UniProt and gene symbol identifiers only and filtered against the matrisome gene list to ensure that at least one interactor of each pair is a matrisome component, for a total of 7367 interactions.
- Basement membraneBASE (https://bmbase.manchester.ac.uk/) (Jayadev et al., 2022). Entries were manually curated to harmonize nomenclature across the dataset (e.g. turning 'native collagen' and 'procollagen' to 'collagen' or turning laminin trimers into their individual constituents) and checked against the matrisome list for a final total of 3534 interactions.
- The KEGG (https://www.genome.jp/kegg/pathway.html) 'focal adhesion' (hsa04510) and 'ECM–receptor interactions' (hsa04512) pathways (Kanehisa et al., 2023). Gene identifiers were converted to gene symbols using the org.Hs.eg.db package in R and the interactions were run against the matrisome list, which retrieved a combined total of 1179 interactions.
- The OmniPath database was accessed programmatically via the OmnipathR package in R (v3.10.1; https://r.omnipathdb.org/) (Türei et al., 2021), filtered against the key 'is_stimulation==1', with multi-subunit complexes simplified by separating each complex into its binary interactions, before the results were filtered against the matrisome gene list for a total of 1650 interactions.
- STRING (https://string-db.org/), physical subnetwork of the whole protein network only. Data were downloaded for human and mouse (taxid:9606 and taxid:10090, respectively), gene identifiers were converted to gene symbols, and the unique results were filtered against the matrisome gene list for a total of 12,191 interactions.
- BioGRID (https://thebiogrid.org/; version 4.4.230), 'multi-validated physical interaction' subset only. Data were filtered for human and mouse (taxid:9606 and taxid:10090, respectively) and filtered against the matrisome gene list for a total of 2491 interactions.

The individual datasets were merged into MatriComDB, an omnibus database of 26,571 unique interactions involving at least one matrisome partner. We assigned a reliability score to each resource based on the level of experimental validation of each protein interaction (entire database versus only part of it) and validation of the proposed interactions by independent groups in the ECM community. This led us to assign the maximum reliability score of 3 to interactions retrieved from MatrixDB and KEGG, a score of 2 to basement membraneBASE and MatrixDB-IMEx extended, and 1 to all other sources.

## MatriCom algorithm and functionalities
### Overview
The online and offline versions of the MatriCom Shiny application were built with the R Project for Statistical Computing and Shiny language (https://shiny.rstudio.com/) and share a common set of functions and 'logic'. The online version is freely accessible at https://matrinet.shinyapps.io/matricom/, while the offline version can be installed from https://github.

com/Izzilab/MatriCom. The difference between the online and local (offline) versions of the application is in their intended use: the online version is suitable to handle the analysis of small- to medium-size datasets (maximum upload size 1 GB) and for the retrieval and analysis of large, curated OA sample datasets (see below). The offline version does not offer access to the OA sample datasets but allows the analysis of larger datasets (the size limit is dictated by the system memory of the local machine running the application). Both versions source the same functions for the upload and preprocessing of data, the actual analysis, the filtering of the results according to user actions, and the graphical and tabular reporting of the results.

The MatriCom function is at the core of the application. This function, given an input scRNA-Seq dataset and a set of options (see below), interfaces with MatriComDB (see above) and a custom-built exclusion database and runs the following analysis:

1. all genes expressed at a certain minimum level (set by the user, defaults to 1 in the app) by each cell identity are fetched;
2. the percentage of each cell identity expressing each of the genes meeting the previous criterion is calculated, and only genes for which expression levels are equal or greater than a threshold (set by the user, defaults to 30 in the app) are kept;
3. using MatriComDB, all possible interactions between the same and different cell identities (homocellular and heterocellular co-expressions, respectively) are identified.

Upon completion of the analysis, the results are enriched with information about matrisome classifications using the matrisome lists implemented in Matrisome AnalyzeR (Petrov et al., 2023). Last, filters can be applied to threshold the results (see below). The output includes graphical and tabular reports as well as analyses aiming to (1) identify the most important communication nodes ('network influencers') and (2) evaluate the enrichment of matrisome elements recently reported to constitute proteomics signature of diseases ('Enrichment Analysis').

### Data input
#### User dataset and input file format stipulation
The intended file format to be used with MatriCom is the Seurat standard for scRNA-Seq data (Satija et al., 2015). Both the online and local versions of the application can internally convert the 'AnnData' format (.h5ad) to a suitable Seurat version. Upon launching, the MatriCom function will represent the Seurat object as version 3 (eventually converting from the current version 5) and search for available assays into the 'data' (normalized counts) slot first, then into the 'counts' slot, then eventually into the slots meant for integration of multiple datasets (typically, 'SCT' and 'integrated'). MatriCom can also process datasets generated through the SCE pipeline, which will be of use to researchers processing their data via the Bioconductor package.

As customary, MatriCom users will be expected to have preprocessed their data for quality assessment and cell type identification. We also recommend that users provide objects with log- or otherwise normalized data to reduce inter-cellular bias.

Importantly, MatriCom implements a strict session-specific data policy: data uploaded by users are neither stored in our server nor can they leak through sessions. User data are purged upon user disconnection or at session timeout.

#### OA sample datasets
Large OA scRNA-Seq datasets have been processed through MatriCom to offer users a streamlined gateway to matrisome communication atlases. In the current release, we offer an interface to Tabula Sapiens (The Tabula Sapiens Consortium et al., 2022), THPA (Karlsson et al., 2021) and additional OA datasets retrieved from Azimuth (Hao et al., 2021) (Table S2). For Tabula Sapiens, we downloaded the full dataset ('Tabula Sapiens – All Cells') available through the CZ Biohub, CZI Single-Cell Biology portal (https://cellxgene.cziscience.com/collections). For THPA, the full dataset ('rna_single_cell_read_count.zip') was downloaded from the THPA portal (https://www.proteinatlas.org/about/download). Additional OA datasets were downloaded from the 'References for scRNA-seq Queries'

section of Azimuth (https://azimuth.hubmapconsortium.org/), focusing on human tissues and organs that could easily be integrated with the larger atlases above, namely bone marrow, kidney, lung (V1) and peripheral blood mononuclear cells. In all cases, data were imported with original cell and tissue/organ identities provided in the respective publications. At data upload, metadata columns become available for the users to choose cell identity labels.

To facilitate the comparison of MatriCom outputs across datasets, we are offering users the possibility of having cell populations within their datasets reannotated using the Census package (Ghaddar and De, 2023). We have also used this package to reannotate cell identities in THPA datasets and other OA datasets based on those present in Tabula Sapiens. This resulted in the reannotation of 22 out of 30 THPA datasets (with brain, bronchus, esophagus, fallopian tube, ovary, placenta, spleen and stomach having no matching samples in Tabula Sapiens) and the four other OA datasets available in MatriCom.

### Output
Upon completion of the analysis, MatriCom returns a 'Global Communication Cluster Map' in the form of a bubble plot reporting the number of communication pairs for each population pair and the full dataset in tabular format (see Results).

Additional outputs from MatriCom include compartmental position and topological and functional enrichments. For compartmental positions, all non-matrisome components are annotated as part of the 'surfaceome' (Bausch-Fluck et al., 2018) or as 'extracellular' or 'intracellular' according to Gene Ontology annotations (The Gene Ontology Consortium, 2019).

At the topological level, we identify influential nodes by converting the results into an acyclic undirected graph and calculating the degree of each node, then subsetting to the nodes with the highest degree (maximum 100, or all the available nodes if the network size is less than 100) and fetching all their first-degree neighbors (i.e. the targets). Then, the contribution (i.e. the influence) of each influencer to the target is calculated as the degree of the influencer divided by the sum of the degrees of all the influencers impinging on the same target and reported as the relative fraction.

At the functional level, we calculate enrichment for 29 matrisome signatures derived from experimental proteomic studies of ECM-enriched samples from a plethora of preclinical and clinical samples previously reported by the Naba laboratory. These signatures are available via the Matrisome Project website (https://matrisome.org) and the Molecular Signature Database (MsigDB; https://www.gsea-msigdb.org/gsea/msigdb) (Liberzon et al., 2011). To calculate the enrichment values, all the matrisome genes found per cell pair in the results are collated and tested against the total number of genes in the matrisome list using a hypergeometric distribution test.

### User experience
To facilitate the use of MatriCom, we have developed an interactive guided tour accessible from the MatriCom home page. The user interface (UI) also contains multiple help buttons that trigger pop-up windows to assist users in choosing the right options for their analysis. Users in need of support or encountering technical difficulties, can reach us using the 'Contact Us' button located on the MatriCom home page.

### Maintenance
We will append the content of MatriComDB periodically with new validated interactions involving at least one matrisome component. We will also strive to periodically expand the list of sample datasets as they become available from large sequencing consortia. In particular, future content expansions will aim to include datasets representative of different disease states.

### Case study: building the kidney matrisome interaction network using MatriCom
We imported the 'Kidney' sample dataset (Stewart et al., 2019) to MatriCom with original annotations, ran analysis using default query parameters, and exported tabular data with all default settings and post-run filters active (Fig. 2A; Table S3).

The code written to perform this analysis is available at https://github.com/Izzilab/MatriCom-analyses/tree/main/CS.

### Relative contribution of populations to the MatriCom communication network
For each population, we evaluated the contribution of expressions to the MatriCom communication network relative to population size (i.e. cell count) in the original sample:

$$\text{Relative contribution per population} = \frac{\text{Expression frequency}}{\text{Population frequency}}, \quad (1)$$

where 'Expression frequency' is determined by the number of times each population participates in a MatriCom communication as either Population1 or Population2 (Fig. 2E), and 'Population frequency' is the proportion of cells per population in the original scRNA-Seq dataset. For each population pair (non-directional), we evaluated the contribution of communication pairs to the MatriCom communication network relative to the size of both populations in the original sample:

$$\text{Relative contribution per population pair} =$$
$$\frac{\text{Communication frequency}}{\text{Fibroblast frequency}} + \frac{\text{Communication frequency}}{\text{Partner frequency}}, \quad (2)$$

where 'Communication frequency' is the combined proportion of communications in the MatriCom output table established by each Fibroblast–Partner and Partner–Fibroblast pair set, and 'Fibroblast frequency' and 'Partner frequency' are the proportions of each population in the original scRNA-Seq dataset. To determine the relative contribution of pairs to the Matrisome–Matrisome and Non-matrisome–Matrisome subnetworks, communication frequencies and proportions per matrisome division and/or category pair were computed based on the total number of communications per subnetwork.

### Fibroblast ECM–receptor communications
We identified all gene pairs in the fibroblast-specific MatriCom communication network that are represented in the KEGG ECM–receptor interactions list (hsa04512), extracted the subset of ECM–receptor pairs for which fibroblasts are the 'sender' population (i.e. express the ligand gene), and determined the proportion of communications involving each unique ligand or receptor gene. Relative contribution of communications to the fibroblast ECM–receptor network per fibroblast–receiver population pair was determined, as described above.

#### Fibroblast collagen VI receptors
From the subset of fibroblast ECM–receptor communications involving one of three collagen VI ligand genes – *COL6A1*, *COL6A2* or *COL6A3* – we extrapolated unique interactions between collagen VI [α1(VI)α2(VI)α3(VI)] trimers and non-integrin cell surface receptors by removing pairs including an integrin receptor gene, concatenating the three collagen VI gene symbols into a shared trimer label and filtering duplicate pairs. For integrin receptors, we separated each collagen VI trimer–integrin heterodimer complex in the KEGG ECM–receptor reference list into its six binary interactions. For each set of integrin genes that encodes a functional heterodimer (e.g. *ITGA1*–*ITGB1*), we screened the fibroblast collagen VI–receptor network for gene pairs established between either the α or β subunit gene and one of the three collagen VI α-chain genes. Multi-subunit interactions between [α1(VI)α2(VI)α3(VI)] trimers and integrin heterodimers are only reported if at least one of the six binary interactions was detected in the fibroblast collagen VI–receptor network.

### Matrisome communication pattern analysis
#### Method
The code written to perform this analysis is available at https://github.com/Izzilab/MatriCom-analyses/tree/main/OA. MatriCom was used to reanalyze open-access scRNA-Seq datasets from Tabula Sapiens (original annotations) and THPA (Census annotation; see above). Tabula Sapiens, having greater depth, was chosen as the primary discovery cohort, with THPA datasets being used to validate findings. Pairs discovered in at least 50% of the available tissues from Tabula Sapiens and present in THPA datasets were termed 'matrisome communication patterns'. A pair-by-tissue similarity matrix was calculated using Pearson correlation, and exemplary

clusters were identified by visual inspection and greedy modularity optimization of the correlation matrix after thresholding for strong correlation (Pearson $R>0.7$) only.

## Enrichment analysis

Enrichment analysis was performed on matrisome co-expression patterns using gprofiler2. Significant results ($P<0.05$) were further filtered to focus on results from the Reactome database (Milacic et al., 2024) and reported in bubble plots in which the diameter of each bubble is proportional to the antilogarithm base 10 ($-\log10$) of the enrichment $P$-value.

## Acknowledgements
We thank the members of the Izzi and Naba laboratories for their feedback on MatriCom and the manuscript. We also thank Professor Sylvie Ricard-Blum for feedback on ECM interactions and the construction of MatriComDB.

## Competing interests
A.N. holds consulting agreements with AbbVie, RA Capital and XM Therapeutics, and receives research support from Boehringer-Ingelheim for work not related to the present study. The other authors declare no competing or financial interests.

## Author contributions
Conceptualization: A.N., V.I.; Data curation: R.L., A.N.; Formal analysis: R.L., A.M.P., P.B.P., A.N., V.I.; Funding acquisition: A.N., V.I.; Investigation: A.M.P., A.N.; Methodology: R.L., A.M.P., P.B.P., A.N., V.I.; Project administration: A.N., V.I.; Resources: A.N., V.I.; Software: R.L., A.M.P., P.B.P., V.I.; Supervision: A.N., V.I.; Visualization: R.L., A.M.P., A.N., V.I.; Writing – original draft: R.L., A.M.P., P.B.P., A.N., V.I.

## Funding
This work was supported in part by the National Human Genome Research Institute (NHGRI) and the National Institutes of Health Common Fund through the Office of Strategic Coordination/NIH Office of the Director [U01HG012680 to A.N.], the National Cancer Institute [R21CA261642 to A.N.], and a start-up fund from the Department of Physiology and Biophysics of the University of Illinois at Chicago to A.N. This research is connected to the DigiHealth-project, a strategic profiling project at the University of Oulu [to V.I.] and the Infotech Institute [to V.I. and P.B.P.]. The project is supported by the Research Council of Finland [DECISION 326291 to V.I.], Syöpäsäätiö [to V.I.], the European Union CARES project [to V.I.] and the Finnish Cancer Institute [K. Albin Johansson Cancer Research Fellowship fund to V.I.]. Open Access funding provided by National Institutes of Health. Deposited in PMC for immediate release.

## Data and resource availability
The web-based MatriCom is deployed as a Shiny Application and is available at https://matrinet.shinyapps.io/matricom/. The MatriCom code package is available at https://github.com/Izzilab/MatriCom. MatriComDB is available at https://github.com/Izzilab/MatriCom/blob/main/inst/webApp/www/MatricomDB. Codes used for the analysis of OA datasets are available at https://github.com/Izzilab/MatriCom-analyses. All other relevant data can be found within the article and its supplementary information.

## Peer review history
The peer review history is available online at https://journals.biologists.com/jcs/lookup/doi/10.1242/jcs.263927.reviewer-comments.pdf

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
