## [Peer Review File · Journal of Cell Science]

MatriCom, a single-cell RNA-sequencing data mining tool to infer cell-extracellular matrix interactions

Rijuta Lamba, Asia M. Paguntalan, Petar B. Petrov, Alexandra Naba and Valerio Izzi
DOI: 10.1242/jcs.263927

Editor: Andrew Ewald

Review timeline

Original submission:	13 February 2025
Editorial decision:	25 April 2025
First revision received:	1 May 2025
Accepted:	4 June 2025

Original submission

First decision letter

MS ID#: jcs.263927

MS TITLE: MatriCom: a scRNA-Seq data mining tool to infer ECM-ECM and cell-ECM communication systems

AUTHORS: Rijuta Lamba; Asia M. Paguntalan; Petar B. Petrov; Alexandra Naba; Valerio Izzi
ARTICLE TYPE: Tools and Resources

Dear Dr Naba,

We have now reached a decision on the above manuscript.

To see the reviewers' reports and a copy of this decision letter, please go to:

As you will see, the reviewers gave favourable reports but raised some critical points that will require amendments to your manuscript. Both reviewers found that you have identified and filled a significant unmet computational need. Both tried the app and found that it worked well. Both reviewers have suggestions for possible changes and improvements to increase the scope and impact of this tool. Please consider and respond to each of their points. I hope that you will be able to carry these out because I would like to be able to accept your paper, depending on further comments from reviewers.

Reviewer 1

Advance summary and potential significance to field

The authors built MatriComDB, of over 25,000 curated interactions involving matrisome components and MatriCom, a web application (<https://matrinet.shinyapps.io/matricom>) and a companion R package, devised to infer communications between ECM components and between different cell populations and the ECM from scRNA-Seq datasets. This is a huge amount of work and fills a current void for the analysis of interactions between ECM components from scRNAseq datasets. The R package is as downloaded from the github repo easy to use and well documented. It was not possible to access and thus evaluate the the Shiny app which is my biggest criticism of the work. Overall, I recommend accepting the article provided that the Shiny app also checks out.

Comments for the author

Working from a Seurat obj is a pain for those who use predominately Bioconductor packages for analysis. I get it, but worry that you are limiting further development using your tools by binding yourself to that structure. This does not reflect on the manuscript or its value as a publication. It is merely something that should be considered by the authors.

Reviewer 2*Advance summary and potential significance to field*

This is a well-written, comprehensive manuscript describing a much-needed research tool for the systematic evaluation of ECM interactions in single-cell datasets (MatriCom) and its associated curated resource of ECM interactions (MatriCom DB). The authors not only follow a high standard for data open-access availability and code transparency, but also make the tool easy-to-use with a great web interface. I tested both the web version and offline version of the MatriCom on my own single-cell data, and both versions performed flawlessly as described. Lastly, the authors used their tool on a specific case (kidney) and on a broad analysis of all organs to show large-scale patterns in putative ECM interactions across tissues --- these foundational analyses and data can also be considered a valuable resource for the research community. In conclusion, this is an excellent manuscript, tool and resource that should be accepted for publication so that the scientific community can benefit. I have only very minor suggestions for the authors to improve the tool in the future.

Comments for the author

- While the REACTOME database is used for the enrichment analysis portion of the workflow, I am curious as to why it was not itself used to build the MatriCom DB as it contains detailed ECM interactions within the Extracellular Matrix Organization category of pathways.
- It would be useful if the tool could perform differential ECM interaction analyses, identifying significantly up- or down-regulated interactions between chosen cell types or categories, since this would increase the hypothesis-testing use-cases for this tool by researchers.

First revisionAuthor response to reviewers' comments**Reviewer 1:****Summary of the advance made in this paper and its potential significance to the field**

The authors built MatriComDB, of over 25,000 curated interactions involving matrixome components and MatriCom, a web application (<https://matrinet.shinyapps.io/matricom>) and a companion R package, devised to infer communications between ECM components and between different cell populations and the ECM from scRNA-Seq datasets. This is a huge amount of work and fills a current void for the analysis of interactions between ECM components from scRNAseq datasets. The R package is as downloaded from the github repo easy to use and well documented. It was not possible to access and thus evaluate the the Shiny app which is my biggest criticism of the work. Overall, I recommend accepting the article provided that the Shiny app also checks out.

> We thank the reviewer for appreciating the effort it took to build MatriComDB and MatriCom and for their positive assessment of our manuscript.

We are somewhat surprised to hear that the online app was not reachable, since we received feedback from Reviewer 2 (*see below*) and other users that the online app performed as expected. We can only assume that it was an unfortunate downtime on the host server side. To the best of our knowledge, the app is always reachable at the address stated in the text (<https://matrinet.shinyapps.io/matricom/>) - although it might take a few seconds to start if the

app was in sleep mode. The host we use (shinyapp.io) is also SSL-compliant and, thus, should be largely compatible with any institutional firewalls.

Of note, users in need of support or encountering technical difficulties, can reach us using the “Contact Us” button located on the MatriCom home page. We have now indicated this in the manuscript (see page 10).

Suggestions to authors

Working from a Seurat obj is a pain for those who use predominately Bioconductor packages for analysis. I get it, but worry that you are limiting further development using your tools by binding yourself to that structure. This does not reflect on the manuscript or its value as a publication. It is merely something that should be considered by the authors.

> Thank you for this suggestion. To address the reviewer’s comment, we have now added a functionality in MatriCom to support, in addition to Seurat objects (.rds and QS) and Python objects (.h5ad), Single Cell Experiment (.sce) file, the predominant format used by the Bioconductor community. We have now added this to the revised manuscript (see pages 9 and 14).

Reviewer 2:

Summary of the advance made in this paper and its potential significance to the field

This is a well-written, comprehensive manuscript describing a much-needed research tool for the systematic evaluation of ECM interactions in single-cell datasets (MatriCom) and its associated curated resource of ECM interactions (MatriCom DB). The authors not only follow a high standard for data open-access availability and code transparency, but also make the tool easy-to-use with a great web interface. I tested both the web version and offline version of the MatriCom on my own single-cell data, and both versions performed flawlessly as described. Lastly, the authors used their tool on a specific case (kidney) and on a broad analysis of all organs to show large-scale patterns in putative ECM interactions across tissues --- these foundational analyses and data can also be considered a valuable resource for the research community.

In conclusion, this is an excellent manuscript, tool and resource that should be accepted for publication so that the scientific community can benefit. I have only very minor suggestions for the authors to improve the tool in the future.

> We thank the reviewer for their enthusiastic assessment of our manuscript and for appreciating the value of MatriCom.

Suggestions to authors

- While the REACTOME database is used for the enrichment analysis portion of the workflow, I am curious as to why it was not itself used to build the MatriCom DB as it contains detailed ECM interactions within the Extracellular Matrix Organization category of pathways.

> We thank the reviewer for this suggestion. We initially intended to add elements of the REACTOME database to enrich MatriComDB. However, we did not see any significant gain in terms of matrisome-matrisome and cell-matrisome interactions over the sources that were already included (especially when the database was “linearized” to avoid logical duplicates). As we plan to update MatriCom DB regularly (at least once per year), we will continue to evaluate sourcing curated interactions from additional resources, including REACTOME. We have now commented on this in the revised manuscript (see page 22).

- It would be useful if the tool could perform differential ECM interaction analyses, identifying significantly up- or down-regulated interactions between chosen cell types or categories, since this would increase the hypothesis-testing use-cases for this tool by researchers.

> Thank you for this suggestion. Multiple types of differential analyses can be envisioned. For example, users may be interested to compare the number of type of interactions per different cell population pairs, which is a functionality readily available in MatriCom as we demonstrated in our case study. Users may also be interested in comparing ECM interaction networks across experimental conditions (e.g., healthy vs diseased tissues). For now, users can perform this type of differential analysis “offline”, by comparing the output of MatriCom run on different sample types. Last, we also offer a differential test for several ECM signatures that have been shown to associate with different disease states, and that is accessible in the “ENRICHMENT ANALYSIS” tab upon successful completion of a MatriCom run.

We have considered implementing more structural statistical testing, which may be what the reviewer is also referring to, however, for now, they would be limited to differential interaction *counts* rather than differential interaction *types* (more likely to hint at function). This is because, as of today, there is limited annotations on the *functional* nature of the interactions reported in MatriComDB. To overcome this limitation, we are currently working with the Gene Ontology consortium to develop rigorous functional ECM ontologies (“biological processes” and “molecular functions”). Once these ontologies are defined, we will first seek feedback from the ECM community prior to publishing them broadly and eventually implementing them in the computational tools we have developed, including MatriCom.

Second decision letter

MS ID#: jcs.263927R1

MS Title: MatriCom: a scRNA-Seq data mining tool to infer ECM-ECM and cell-ECM communication systems

Authors: Rijuta Lamba; Asia M. Paguntalan; Petar B. Petrov; Alexandra Naba; Valerio Izzi
Article Type: Tools and Resources

Dear Dr Naba,

I am happy to tell you that your manuscript has been accepted for publication in Journal of Cell Science, pending standard publication integrity checks.

Reviewer 1

Advance summary and potential significance to field

See previous review.

Comments for the author

Reviewer 2

Advance summary and potential significance to field

As before.

Comments for the author